# The impact of pulmonary rehabilitation on sleep quality in patients with chronic obstructive pulmonary disease: A systematic review and meta-analysis

Shuiyan Dai[1]*, Chun Shing Kwok[2]

**1** The Second People's Hospital of Foshan, Foshan, China, **2** Department of Cardiology, Leighton Hospital, Mid Cheshire Hospitals NHS Foundation Trust, Crewe, United Kingdom

* Shuiyan.Dai@mail.bcu.ac.uk

## Abstract

### Background

Pulmonary rehabilitation (PR) is a significant component of chronic obstructive pulmonary disease (COPD) management, but whether it has any impact on patient sleep quality is unknown.

### Methods

We conducted a systematic review of the literature to evaluate the impact of PR on sleep quality in patients with COPD. Searches of the MEDLINE, EMBASE and Cochrane Library databases were performed, and data were extracted from relevant studies. Meta-analysis with the random effects model was performed to determine if PR is associated with any difference in the Pittsburgh Sleep Quality Index (PSQI).

### Results

A total of 16 studies were included in this review with 1,478 patients, of which 1,169 took part in PR. The PR programmes were variable, some being 8-week programmes while others were 12-week, and some lasted for 6 months. The pooled results of 372 patients suggest that PR is associated with a significant improvement in the PSQI (mean difference 2.33 95% CI 0.46 to 4.20, $I^2$ = 93%, 6 studies). Removal of one study, significantly reduced the statistical heterogeneity and the effect size (mean difference 0.91 95% CI 0.35–1.48, $I^2$ = 0%, 5 studies). For other outcomes such as sleep onset latency, wake after sleep onset, total sleep time, sleep efficiency, and number of awakenings, there were no consistencies to suggest any benefit associated with PR.

**Data availability statement:** All relevant data are within the paper.

**Funding:** The author(s) received no specific funding for this work.

**Competing interests:** The authors have declared that no competing interests exist.

## Conclusions

PR appears to be associated with improvements in sleep quality in patients with COPD. More studies are needed to determine what the ideal PR programme is that best improves sleep quality and whether certain patients benefit more compared to others.

## Introduction

Chronic obstructive pulmonary disease (COPD), characterised by persistent airflow limitations and relevant symptoms, is a chronic airway disease that is common, preventable, and treatable [1]. COPD has a global mortality rate of 5.8% [2] and there are over 390 million sufferers of the condition [3]. The impact of COPD on the quality of life of patients is significant as anxiety and depression ranges from 10–55% among inpatients and 13–46% among outpatients [4]. Despite efficacious treatments and comprehensive guidelines for patient care, moderate-to-very severe COPD represents a major economic burden for healthcare providers [5].

Sleep quality is poor in patients with severe COPD compared with normative populations of similar age, and daytime hypoxaemia is independently associated with impaired sleep efficiency [6]. Approximately 70% of COPD patients typically complain of difficulty sleeping, two to three times more than the general population [7]. One study suggests that 39% of patients with nocturnal cough or wheeze report difficulty initiating or maintaining sleep [8]. Patients with COPD may experience anxiety, depression, shortness of breath and coughing, sputum, and hypoxaemia during the night [9,10]. There is also some evidence to suggest that patients with COPD may be affected by an overlap with obstructive sleep apnoea as the prevalence of obstructive sleep apnoea ranges from 10 to 65% [11]. There appears to be a bidirectional relationship between sleep quality and clinical outcomes, with sleep disturbances leading to systemic inflammation, immune impairment, lack of exercise and cognitive changes, which may affect medication adherence and lead to poor clinical outcomes, nocturnal COPD symptoms, and reduced physical activity [12]. Sleep disturbance in patients with COPD likely contributes to the non-specific daytime symptoms of chronic fatigue, lethargy and overall impairment in quality of life [13].

Pulmonary rehabilitation (PR) is a fundamental component of non-pharmacological COPD treatment. PR typically consists of two to three weekly sessions of exercise training including aerobic and strength training for eight to twelve weeks and may include education and behaviour change interventions [14]. It is effective in improving daily symptoms such as dyspnoea, reducing the frequency of hospitalisation, and improving the ability of COPD patients to perform activities associated with daily living [15,16]. PR is also beneficial in reducing anxiety and depression in COPD patients [17], which may also have an impact on sleep quality. Whether PR has any direct impact on sleep in patients with COPD is controversial as some studies have demonstrated that PR improves sleep quality in people with COPD [18–20] while

other studies found no difference in sleep quality [21–23]. Therefore, we conducted this systematic review of the literature to determine if PR has any direct impact on sleep quality in patients with COPD.

## Methods

This systematic review and meta-analysis were conducted and reported according to the Preferred Reporting Item of the Guidelines for Systematic Reviews and Meta-Analysis (PRISMA) [24]. The systematic review was registered on PROSPERO [CRD42023403543].

### Literature search

We used the OVID platform to search for relevant studies on MEDLINE and EMBASE in February 2023 and in the Cochrane Library database in March 2023. The specific search terms were (pulmonary rehabilitat*) AND (sleep) AND (chronic obstructive pulmonary disease OR COPD OR chronic bronchitis OR emphysema).

### Study inclusion and exclusion criteria

Studies were included if they evaluated sleep quality in people with COPD who underwent pulmonary rehabilitation and there was no restriction based on the presence or type of comparator group. The participants were patients with a diagnosis of COPD. The intervention is pulmonary rehabilitation and the comparator is the same patient prior to taking part in pulmonary rehabilitation. The outcome is sleep quality. There was no restriction on study design which could be observational or clinical trial. However, studies which included adults were included. Studies had to have original data which could be in full manuscript or abstract form so letters, editorials, comments, and reviews without original data were excluded. The reference lists of potentially relevant studies were checked for additional studies.

### Study selection, data extraction and quality assessment

The screening of articles was done independently by two reviewers (SD and CSK). Both reviewers (SD and CSK) collected data on study design, country, year, number of patients, mean age, proportion of men and study inclusion criteria. In addition, definitions of PR, definitions of sleep quality, follow up and result, and concordance to pulmonary rehabilitation were also collected. Where there were disagreements in study selection, data extraction and quality assessment between the two reviewers, the third reviewer (SS) was consulted and a decision was made by consensus. Study quality assessments for observational studies were carried out using the Ottawa-Newcastle Scale for observational studies [25]. Stars were awarded, out of a maximum possible score of 9 stars, based on whether a study was representative of a general cohort of patients with COPD, involved the selection of a control group (cohort prior to PR), ascertainment of PR, demonstration that sleep quality was assessed at baseline, comparability of the cohort, ascertainment of the sleep quality, adequacy of the length of follow up, and information on loss to follow up. The study quality assessment for randomised trials was performed using the Cochrane Risk of Bias Tool [26]. The domains assessed were bias from randomization process, bias due to deviations from intended intervention, bias in measurement of the outcome, and bias in selection of the reported result.

### Data synthesis

RevMan 5.4 (The Nordic Cochrane Centre, The Cochrane Collaboration, Copenhagen, Denmark) was employed in performing meta-analysis with the random effects model using the mean difference method. The primary outcome of this study was the Pittsburgh Sleep Quality Index (PSQI) and the mean and standard deviation in the PSQI before and after PR were pooled. The PSQI was chosen because it is the most commonly used generic measure of sleep quality in clinical and research settings [27]. The main analysis which included studies that were of randomized controlled trial (RCT) and cohort in design was stratified by study design so that the pooled findings from RCT could be distinguished from cohort

studies. Statistical heterogeneity was assessed using the $I^2$ statistic, with $I^2$ values of between 30%-60% representing a moderate degree of heterogeneity [28]. If there were more than 10 studies in the meta-analysis and the statistical heterogeneity was less than 50% [29], we planned to perform an asymmetry test to determine whether there was publication bias. We undertook leave-one-out sensitivity analyses to determine the source of statistical heterogeneity.

## Results

### Description of included studies

The search of MEDLINE and EMBASE yielded 258 potentially relevant records and the search of the Cochrane Trial Register yielded 90 potentially relevant records (Fig 1). A total of 16 studies were included after screening titles and abstracts and reviewing full texts of the relevant studies [15–20,30–39].

As described in Table 1, included in this review were 16 studies which comprised 4 randomized controlled trials, 9 prospective cohort studies, and 3 retrospective cohort studies. Among these studies, there were a total of 1478 patients of which 1169 took part in PR. The average age across 13 studies, that reported mean age, was 66.6 years and the proportion of male patients was 64.1%.

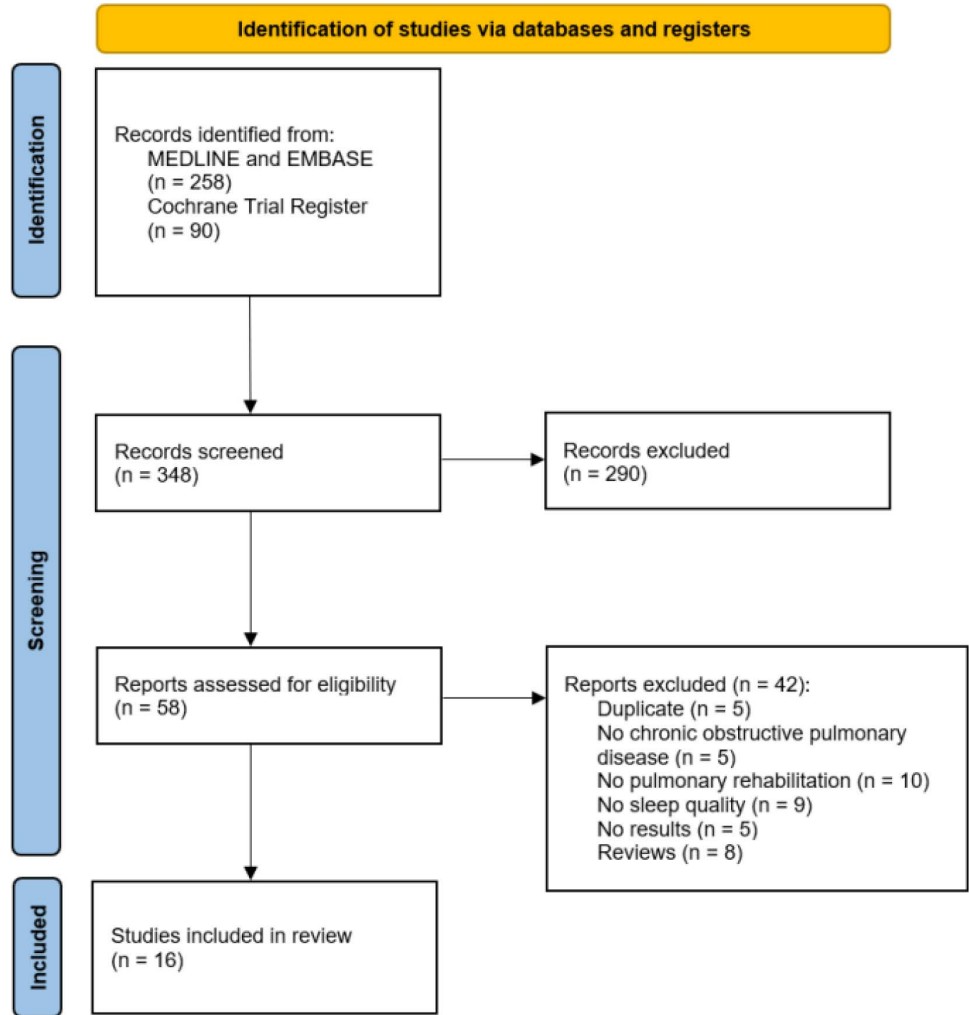

**Fig 1. PRISMA flow diagram of study selection.**

**Table 1. Study design, participant characteristics and participant inclusion criteria.**

| Study ID | Study design; Country; Year | No. of patients (PR, control) | Mean age | % Male | Participant inclusion criteria |
|---|---|---|---|---|---|
| **Benzo 2022** | Randomized clinical trial; United States; recruitment started in 2019. | 375 (188 PR, 187 control) | 69.0 | 43.5% | Patients with a clinical diagnosis of COPD confirmed by records, age 40 years or older, a history of a minimum of 10 pack-years of smoking, and the ability to communicate English who were randomized to remote patient monitoring with health coaching or usual care. |
| **Cox 2019** | Post-hoc analysis of RCT; Australia; 2011–2015. | 48 (all PR) | 70.0 | 43.8% | Patients had a diagnosis of COPD, a smoking history ≥10 pack years, no respiratory exacerbations within the previous four weeks, and absence of comorbidities that preclude exercise training. |
| **Grosbois 1996** | Prospective cohort study; France; 1990–1995. | 88 (all PR) | 62.1 | 87.5% | Patients had a diagnosis of COPD, respiratory and cardiac stability with no cardiovascular contraindications or no respiratory difficulties or no restrictions to daily activities. |
| **Jin 2021** | Randomized clinical trial; China; 2019 to 2020. | 90 (30 PR 3x weekly, 30 PR 5x weekly, 30 control) | 45.9 | 54.4% | Patients aged 35 to 75 years were diagnosed with COPD, were able to be discharged under stable conditions, and cooperated well with this study. |
| **Kittah 2015** | Prospective cohort study; United States; 2012 to 2014. | 17 (all PR) | – | – | Patients with COPD referred for pulmonary rehabilitation. |
| **Lan 2014** | Prospective cohort study; China; 2011 to 2012. | 34 (all PR) | 70.2 | 82.4% | Patients had a diagnosis of COPD based on the GOLD guidelines and had stability with no exacerbations or worsening of respiratory symptoms, no increased use of rescue medication, no unscheduled visits for at least 3 months, and the ability to mobilize independently. |
| **Mak 2022** | Retrospective cohort study; United States; 2013 to 2020. | 98 (all PR) | 72 | 100% | Patients who were veterans with COPD who completed pulmonary rehabilitation. |
| **McDonnell 2014** | Prospective cohort study; England; 2011 to 2012. | 52 (28 PR, 24 control) | 70.4 | 46.1% | Patients had a diagnosis of COPD based on spirometry. |
| **Nobeschi 2017** | Prospective cohort study; Brazil; Unclear. | 18 (all PR) | – | – | Patients with moderate COPD who did not undergo treatment for sleep disorder and who quit smoking. |
| **Roberts 2018** | Retrospective cohort study; Australia; 2012 to 2017. | 329 (all PR) | 69.5 | 52% | Patients with COPD who completed pulmonary rehabilitation. |
| **Shah 2013** | Prospective cohort study; United States; Unclear. | 10 (all PR) | – | – | Patients with COPD who underwent pulmonary rehabilitation. |
| **Silva 2018** | Prospective cohort study; Brazil; Unclear. | 33 (12 elastic band, 11 elastic tubes, 10 weight training) | 68.6 | 60.6% | Patients had a diagnosis of COPD. |
| **Soler 2013** | Prospective cohort study; United States; 2008. | 64 (all PR) | 68.0 | 48.4% | Patients were adults aged 18 years or older with a diagnosis of chronic lung disease confirmed by medical history, physical examination, and pulmonary function tests, and clinically stable with no recent acute respiratory exacerbation, symptoms, and no other medical conditions that would interfere with full participation in the program. |
| **Spielmanns 2022** | Randomized clinical trial; Germany; 2019–2021. | 67 (33 PR, 34 control) | 64.3 | 50.7% | Patients had a diagnosis of COPD with GOLD stage II to IV. |

*(Continued)*

**Table 1.** (Continued)

| Study ID | Study design; Country; Year | No. of patients (PR, control) | Mean age | % Male | Participant inclusion criteria |
|---|---|---|---|---|---|
| Thapamagar 2021 | Retrospective cohort study; United States; 2012 to 2018. | 69 (all PR) | 69 | 97% | Patients were Veterans aged 18 years or older with a diagnosis of COPD. |
| Yu 2022 | Randomized clinical trial; China; 2020 to 2021. | 86 (43 PR, 43 control) | 66.9 | 66.3% | Patients with a diagnosis of COPD with GOLD stage I to III who together with their families gave informed consent and were in a stable condition with treatment and had the ability think and communicate. |

COPD=chronic obstructive pulmonary disease, RCT=randomized controlled trial, PR=pulmonary rehabilitation.

## Study quality assessment

The quality assessments of the included studies are shown in Tables 2 and 3. Among the eleven observational studies, six had seven out of nine stars and would be considered as high quality compared to the other five studies which had six out of nine stars based on the Ottawa-Newcastle Scale. Among the five randomized controlled trials included in the review, two studies were classified as low risk for all five domains of the Cochrane risk of bias tool.

## PR and sleep quality results from included studies

The descriptions of the results from the studies included in the analysis are shown in Table 4. The PR programmes were variable, some were 8-weeks in length while others were 12-weeks or 12 months. Some involved health coaching [27] while others compared home and centre-based PR [18]. Silva et al compared elastic band training and elastic tube training to that using conventional weight machines [20]. The PR in the study by Soler et al included psychologist led support with the exercise programme [16] and Spielmanns et al included an app that helped deliver the physical exercise training sessions [26]. Yu and Fan included a mindfulness behavioural intervention in their PR [29]. However, there was no consistent measure for sleep quality other than the PSQI which was reported in 8 studies, and one additional study used a modified Chinese version of the PSQI. Other markers of sleep quality included sleep onset latency, wake after sleep onset, sleep wake time, sleep efficiency, number of awakenings, visual analogue scale for sleep, Epworth Sleepiness score, and sleep disturbances.

## PR and sleep quality as assessed by the PSQI

A total of 6 studies were included in the pooled analysis of PR on sleep quality as assessed by the PSQI (Fig 2). A total of 372 patients underwent PR. The pooled results suggest that PR is associated with a significant improvement in sleep quality (mean difference 2.33 95% CI 0.46 to 4.20, $I^2$ = 93%, 6 studies). The benefit of PR was observed to a greater extent among studies that were of RCT design (mean difference 3.95 95% CI 0.36 to 7.55, $I^2$ = 97%, 2 studies) compared to cohort studies (mean difference 0.92 95% CI 0.12 to 1.71, $I^2$ = 0%, 4 studies). However, the source of the heterogeneity was the study by Jin et al [17]. If this study is removed the estimate would still be significantly different but the effect size would be smaller with no statistical heterogeneity (mean difference 0.91 95%CI 0.35–1.48, $I^2$ = 0%, 5 studies).

## Results from studies that were not in the pooled analysis

Cox et al. reported no significant difference in sleep onset latency ($p$ = 0.2), wake after sleep onset ($p$ = 0.7), total sleep time ($p$ = 0.07), sleep efficiency ($p$ = 0.2), and the number of awakenings ($p$ = 0.4) with PR [18]. Grosbois et al reported that sleep quality, based on a visual analogue scale, was significantly improved after PR (p < 0.001) [30]. Kittah et al

**Table 2. Study quality assessment by Newcastle-Ottawa scale.**

| Study ID | Representativeness of cohort (*) | Reliable selection of non-exposed cohort (*) | Reliable ascertainment of exposure (*) | Demonstration that outcome was not present at start of study (*) | Comparability (**) | Reliable assessment of outcome (*) | Reliable assessment of follow up (*) | Reliable adequacy of follow up | Total stars |
|---|---|---|---|---|---|---|---|---|---|
| Grosbois 1996 | Yes, patients with COPD, respiratory and cardiac stability with no cardiovascular contraindications and no respiratory difficulties or no restrictions to daily activities. | Yes, patients in the same condition before PR. | Yes, patients received PR monitored using the Nellcor N200 pulse. | Yes, patient outcome was change in sleep quality based on Visual analogue scale from baseline. | No, unadjusted results and all patients received PR. | Yes, sleep quality was based on the Visual analogue scale. | Yes, assessment after intervention. | Yes, no loss to follow up reported. | 7 |
| Kittah 2015 | Yes, patients with COPD referred for pulmonary rehabilitation. | Yes, patients in the same condition before PR. | Yes, patients received PR. | Yes, patient outcome was change in sleep quality based on PSQI, Epworth Sleepiness Scale and actigraphy from baseline. | No, unadjusted results and all patients received PR. | Yes, sleep quality was based on the PSQI, Epworth Sleepiness Scale and actigraphy. | Yes, 9 weeks follow up. | No, 9 out of 26 did not complete PR or were excluded. | 6 |
| Lan 2014 | Yes, patients with COPD based on the GOLD guidelines and had stability with no exacerbations or worsening of respiratory symptoms, no increased use of rescue medication, no unscheduled visits for at least 3 months, and the ability to mobilize independently. | Yes, patients in the same condition before PR. | Yes, patients received PR monitored by rehabilitation therapist. | Yes, patient outcome was change in sleep quality based on Chinese version of PSQI from baseline. | No, unadjusted results and all patients received PR. | Yes, sleep quality based on Chinese version of PSQI. | Yes, 12 weeks follow up. | Yes, no loss to follow up reported. | 7 |
| Mak 2022 | Yes, patients who were veterans with COPD who completed pulmonary rehabilitation. | Yes, patients in the same condition before PR. | Yes, patients received PR. | Yes, patient outcome was change in sleep quality based on PSQI from baseline. | No, unadjusted results and all patients received PR. | Yes, sleep quality was based on the PSQI and actigraphy. | Yes, 12 months follow up. | No, 16 completed 12-month post-PR assessment out of 98 participants enrolled in PR. | 6 |
| McDonnell 2014 | Yes, patients had a diagnosis of COPD based on spirometry. | Yes, patients were allocated to observation or control. | Yes, patients received PR. | Yes, patient outcome was change in sleep quality based on PSQI from baseline. | No, unadjusted results. | Yes, sleep quality was based on the PSQI. | Yes, 8 weeks follow up. | No, never started or dropped out of PR 23 out of 61. Total drop-out or excluded 33. | 6 |
| Nobeschi 2017 | Yes, Patients with moderate COPD who did not undergo treatment for sleep disorder and who quit smoking. | Yes, patients in the same condition before PR. | Yes, patients received PR. | Yes, patient outcome was change in sleep quality based on PSQI, Epworth Sleepiness Scale from baseline. | No, unadjusted results and all patients received PR. | Yes, sleep quality was based on the PSQI, Epworth Sleepiness Scale | Yes, 8 weeks follow up. | Yes, no loss to follow up reported. | 7 |

*(Continued)*

**Table 2.** (Continued)

| Study ID | Representativeness of cohort (*) | Reliable selection of non-exposed cohort (*) | Reliable ascertainment of exposure (*) | Demonstration that outcome was not present at start of study (*) | Comparability (**) | Reliable assessment of outcome (*) | Reliable assessment of follow up (*) | Reliable adequacy of follow up | Total stars |
|---|---|---|---|---|---|---|---|---|---|
| Roberts 2018 | Yes, patients with COPD who completed pulmonary rehabilitation. | Yes, patients in the same condition before PR. | Yes, patients received PR. | Yes, patient outcome was change in sleep quality based on PSQI, Epworth Sleepiness Scale from baseline. | No, unadjusted results and all patients received PR. | Yes, sleep quality was based on the PSQI, Epworth Sleepiness Scale | Yes, assessment after intervention. | Yes, no loss to follow up reported. | 7 |
| Shah 2013 | Yes, patients with COPD who underwent pulmonary rehabilitation. | Yes, patients in the same condition before PR. | Yes, patients received PR. | Yes, patient outcomes were change in sleep variables from baseline. | No, unadjusted results and all patients received PR. | Yes, sleep quality was based on actigraphy. | Yes, assessment after intervention. | No, 2 patients were excluded because they were ill at post-PR actigraphy. | 6 |
| Silva 2018 | Yes, patients had a diagnosis of COPD. | Yes, patients in the same condition before PR. | Yes, patients divided into three groups respectively received elastic band training, elastic tube training and training with conventional weight machines. | Yes, patient outcome was change in sleep quality based on based on Mini-sleep questionnaire from baseline. | No, unadjusted results and all patients received PR. | Yes, sleep quality based on Mini-sleep questionnaire. | Yes, 3 months follow up. | Yes, no loss to follow up reported. | 7 |
| Soler 2013 | Yes, patients aged 18 years or older with COPD and clinically stable with no recent acute respiratory exacerbation, symptoms, and no other medical conditions that would interfere with full participation in the program. | Yes, patients in the same condition before PR. | Yes, patients received PR and psychologist led psychosocial support. | Yes, patient outcomes were change in sleep variables from baseline. | No, unadjusted results and all patients received PR. | Yes, sleep quality based on PSQI global. | Yes, 8 weeks follow up. | Yes, no loss to follow up reported. | 7 |
| Thapamagar 2021 | Yes, patients were Veterans aged 18 years or older with a diagnosis of COPD. | Yes, patients in the same condition before PR. | Yes, patients received PR monitored by Actiwatch 2. | Yes, patient outcomes were change in sleep variables from baseline. | No, unadjusted results and all patients received PR. | Yes, sleep quality was based on the PSQI, Epworth Sleepiness Scale. | Yes, 8 weeks follow up. | No, 15/90 patients dropped out of PR programme or did not have complete actigraphy data. | 6 |

COPD=chronic obstructive pulmonary disease, PSQI=Pittsburgh Sleep Quality Index, PR=pulmonary rehabilitation, MDT=multidisciplinary team.

found that after PR, there were no statistically significant improvements in sleep efficiency ($p = 0.35$), sleep onset latency ($p = 0.23$), wake after sleep onset ($p = 0.51$), and total sleep time ($p = 0.46$) as measured by wrist actigraphy or PSQI ($p = 0.43$) [25]. In the study by Mak et al. [28], it was found that PR worsened the total time in bed and total sleep time or did

**Table 3. Study quality assessment by cochrane risk of bias tool for randomised trials.**

| Study ID | Bias from randomization process | Bias due to deviations from intended interventions | Bias due to missing outcome data | Bias in measurement of the outcome | Bias in selection of the reported result |
|---|---|---|---|---|---|
| Benzo 2022 | Some concern, randomized in a 1:1 ratio on the basis of a pregenerated sequence of assignments through a computer-generated permuted block randomization with blocks of size four, allocation concealment unclear. | Low risk, 12-week program with weekly healthcare calls and a remote monitoring system, Garmin Vivofit activity monitor and oximeter vs usual care and educational packet for weekly self-study. | Low risk, participants in both arms did not complete measures at 3 months, lost contact and refused participation. Some patient died and became ineligible in usual care group. | Some concern, assessors were not blinded to intervention received. | Low risk, trial analysed according with pre-specified plan. |
| Cox 2019 | Low risk, randomisation undertaken using a computer-generated sequence and allocation will be concealed using sealed, opaque envelopes. | Low risk, home versus hospital-based rehabilitation program. | Low risk, no missing outcome data reported. | Low risk, Objective measures of sleep quality were obtained from the SenseWear Armband. | Low risk, trial analysed according with pre-specified plan although this was post-hoc analysis. |
| Jin 2021 | Some concern, random number table used for randomisation but allocation concealment unclear. | Low risk, pulmonary rehabilitation training three times a week in addition to conventional drug treatment vs drug treatment alone. | Low risk, no missing outcome data reported. | Some concern, participants and assessors were not blinded. | Low risk, trial analysed according with pre-specified plan. |
| Spielmanns 2022 | Low risk, randomisation using a software randomizer and fabrication of lists done by the sponsor for allocation concealment. | Low risk, KAIA COPD app program vs usual care. | Low risk, only 6 out of 67 withdrew informed consent or discontinued by investigators. | Low risk, electronic patient-reported outcomes and activity and sleep measured passively and continuously for each participant by the activity tracker. | Low risk, trial analysed according with pre-specified plan. |
| Yu 2022 | Some concern, randomization by random number table but allocation concealment unclear. | Low risk, mindfulness behavioural intervention combined with progressive breathing training vs symptomatic treatment during hospitalization and routine nursing measures in internal medicine. | Low risk, no missing outcome data reported. | Some concern, participants and assessors were not blinded. | Low risk, trial analysed according with pre-specified plan. |

not improve sleep onset latency, sleep efficiency, wakefulness after sleep onset, and PSQI. The study by Nobeschi et al reported that PR was associated with improvements in sleep efficiency and latency but there was no significant difference in daytime sleepiness [32]. Roberts et al found significant reduction in mean PSQI score and Epworth sleepiness scale following PR (p < 0.0001 and $p$ = 0.005, respectively) [33]. The study by Shah et al showed that sleep efficiency, and WASO time significantly improved after PR (p < 0.05 for both outcomes) [31]. The study of elastic band, elastic tube and conventional training found no difference in sleep outcomes for any of the groups ($p$ > 0.05 for all groups) [20]. Soler et al reported a reduction in PSQI score and sleep disturbance post PR but no difference in subjective sleep quality, sleep latency, sleep duration, sleep efficiency, use of sleep medication and daytime dysfunction [16] Spielmanns et al found no difference in duration of sleep or sleep efficiency following PR [26]. There was no significant difference in sleep efficiency ($p$ = 0.32), wakefulness after sleep onset ($p$ = 0.70), number of nightly awakening events ($p$ = 0.77) and mean change in Epworth sleepiness scale ($p$ = 0.54) in the study by Thapamagar et al. [34] Yu and Fan reported that quality of life in the study group was better than control group ($p$ < 0.05) [29].

**Table 4. Definition of pulmonary rehabilitation, definition of sleep quality, predictors of sleep outcome and concordance to pulmonary rehabilitation.**

| Study ID | Definition of pulmonary rehabilitation | Definition of sleep quality | Follow up and result | Concordance to pulmonary rehabilitation |
|---|---|---|---|---|
| Benzo 2022 | 12-week program with weekly health coaching calls, a remote monitoring system with activity monitor and oximeter and computer tablet for daily steps and self-reported symptoms. Individuals were asked to complete 3 exercise practices daily, 6 days a week with 2 walks of at least 6 minutes and upper extremity simple yoga for at least 12 minutes. | PSQI. | PSQI: Intervention group: Baseline 9.39 ± 3.58, at 3 months 8.48 ± 3.26, change in score -0.72 95%CI -1.09 to -0.34.<br>Usual care: Baseline 9.33 ± 3.52, at 3 months 9.13 ± 3.61, change in score 0.03 95%CI -0.36 to 0.42.<br>Difference mean change: -0.75 95%CI -1.28 to -0.21, p = 0.0068 | 45 patients randomized to the pulmonary rehabilitation did not complete: 17 did not complete measure, 11 no baseline measure, 12 refuse to participate, 2 lost contact, 3 disease progression. |
| Cox 2019 | Home vs centre-based PR. Centre-based PR was 8-week, twice-weekly outpatient group supervision programme, including individually prescribed exercise training and self-management education, which will include 30 minutes of aerobic exercise, upper and lower body strength training, functional tasks, and free weight training. Home-based PR had exercise for 8 weeks, five times a week for 30 minutes each time with resistance training and a home diary. | Sleep onset latency, wake after sleep onset, total sleep time, sleep efficiency and number of awakenings based on accelerometer positional data from SenseWear Armband. | Change and effect size of outcomes: sleep onset latency -2.2 95%CI -15 to 5 effective size 0.1 p = 0.2, wake after sleep onset -3 95%CI -21 to 16 effective size 0.04 p = 0.7, total sleep time 8 95%CI -13 to 44 effective size 0.2 p = 0.07, sleep efficiency 1 95%CI -3 to 6 effect size 0.1 p = 0.2, number of awakenings 0 95%CI -1 to 2 effective size 0.08 p = 0.4. No difference in any outcome for completers and non-completers of PR. | 7 out of 48 did not complete PR. |
| Grosbois 1996 | PR with 6 patients supervised by 2 physiotherapists and a pulmonologist for 7 weeks, with 3 2-hour sessions a week. Each session begins with a 30-minute health education programme, traditional respiratory physiotherapy techniques and retraining on a bike. Patients are continuously monitored using the Nellcor N200 pulse oximeter. | Visual analogue scale for sleep. | Before and after PR sleep: 4.7 ± 3.1 vs 5.7 ± 2.4 p < 0.001. | – |
| Jin 2021 | 6-months PR for 3 or 5 times a week with pursed-lip breathing, abdominal breathing and abdominal resistance training for 15–20 min/day. | PSQI. | PSQI before and 6 months after intervention for: control group 16.49 ± 2.99 vs 14.33 ± 3.02 p < 0.05, observation group 3x 16.84 ± 3.03 vs 12.09 ± 2.66 p < 0.05, observation group 5x 16.38 ± 2.37 vs 10.11 ± 2.05 p < 0.05. After 6 months all groups reduced but most for observation group 5x. | Not reported. |
| Kittah 2015 | 9 weeks of PR. | PSQI and Epworth Sleepiness Scale score with actigraphy. | After PR, there was no statistically significant improvements in sleep efficiency (p = 0.35), sleep onset latency (p = 0.23), wake after sleep onset (p = 0.51), and total sleep time (p = 0.46) as measured by wrist actigraphy or PSQI (p = 0.43). | 9 out of 26 did not complete PR or were excluded. |
| Lan 2014 | 12-week hospital based out-patient PR with 2 sessions/week. Each session had formal education and exercise training for ~40 min monitored by rehabilitation therapist. | Chinese version of PSQI with score of >5 indicates poor quality of sleep. | After PR, the mean PSQI decreased from 9.41 ± 4.33 to 7.82 ± 3.90, p < 0.001. In the PSQI, sleep duration, sleep disturbance, and daytime dysfunction showed significant improvement after PR. Before PR, 85.3% had poor quality sleep (PSQI >5) and this decreased to 64.7% after PR. | Not reported. |

*(Continued)*

**Table 4.** (Continued)

| Study ID | Definition of pulmonary rehabilitation | Definition of sleep quality | Follow up and result | Concordance to pulmonary rehabilitation |
|---|---|---|---|---|
| Mak 2021 | 8-week structured PR with 12 months of unstructured exercise programme. | PSQI and total time in bed, total sleep time, sleep onset latency, sleep efficiency, wakefulness after sleep onset, and total nocturnal awakenings. | Objective sleep variables from actigraphy worsened for total time in bed and total sleep time or did not improve for sleep onset latency, sleep efficiency, and wakefulness after sleep onset. PSQI did not improve significantly after 12 months post-PR (mean change, -1.3, 95%CI -3.1 to 0.5, p = 0.14). | 16 completed 12-month post-PR assessment out of 98 participants enrolled in PR. |
| McDonnell 2014 | 8-week community PR programme with at least 1 session a week. | PSQI. | Observation vs control group difference pre- and post-PR: observation 0.79 95%CI -0.35 to 1.93 p = 0.170, control 0.71 95%CI -0.56 to 1.98, p = 0.261. | Never started or dropped out of PR 23 out of 61. Total drop-out or excluded 33. |
| Nobeschi 2017 | PR for 8 weeks but not described in detail. | Sleep quality based on PSQI and Epworth scale. | Baseline sleep quality good 33.3%, poor 44.4%, sleep disturbance 22.3%. Post-PR sleep quality good 55.6%, poor 27.7%, sleep disturbance 16.7%. Significant improvement in sleep efficiency and latency after PR. Daytime sleepiness 72.2%, no difference pre- and post-PR (p > 0.05). | Not reported. |
| Roberts 2018 | PR was not described. | PSQI with poor quality of sleep defined by PSQI > 5 and Epworth Sleepiness Scale. Responders had poor sleep quality at baseline and improved their PSQI by ≥3 units following PR. | Poor quality sleep (PSQI > 5) 219/329. Mean PSQI decreased by 0.95 ± 3.14 units (p < 0.0001) and Epworth Sleepiness Scale decreased by 0.63 ± 4.07 units (p = 0.005). Responders following PR: 88/219 patients | Not reported. |
| Shah 2013 | PR was not described. | Sleep efficiency, wake after sleep onset, total sleep time, number of awakening events, sleep onset latency. | Completed PR had significant improvement in sleep efficiency from 64 ± 16% to 81 ± 6% (p < 0.05). WASO time also improved from 112 ± 44 minutes to 45 ± 24 minutes (p < 0.05). There was a trend towards improvement in total sleep time from 304 ± 104 minutes to 379 ± 83 minutes, though not statistically significant (p = 0.06). The number of awakenings and sleep onset latency were not different. | 2 patients were excluded because they were ill at post-PR actigraphy. |
| Silva 2018 | Elastic band training, elastic tube training and training with conventional weight machines for 3 months. | Sleep disturbances based on Mini-sleep questionnaire. | Change post-PR follow up: elastic band training 1.08 ± 1.20, elastic tube training 0.18 ± 3.28, conventional training 2.00 ± 1.83. Effect size for group 0.02 p = 0.72, group x time 0.08 p = 0.32, time 0.001 p = 0.93. | Not reported. |
| Soler 2013 | 16 sessions over 8 weeks with individual and group education, physical and respiratory care instruction, psychosocial support and exercise training. There was psychologist led psychosocial support and exercise programme with lower and upper extremity endurance, strength and flexibility training. | PSQI global and indices subjective sleep quality, sleep latency, sleep duration, sleep efficiency, sleep disturbances, use of sleep medications and daytime dysfunction. | Pre- vs post- PR for obstructive airway disease: PSQI global 6.6 ± 3.9 vs 5.5 ± 3.6 p < 0.05, subjective sleep quality 0.7 ± 0.7 vs 0.6 ± 0.7 p > 0.05, sleep latency 0.9 ± 1.0 vs 0.8 ± 0.9, p > 0.05, sleep duration 0.9 ± 1.0 vs 0.9 ± 1.1 p > 0.05, sleep efficiency 0.8 ± 1.1 vs 0.7 ± 1.0, p > 0.05, sleep disturbance 1.4 ± 0.6 vs 1.1 ± 0.5 p < 0.01, use of sleep medication 0.8 ± 1.2 vs 0.6 ± 1.1 p > 0.05, daytime dysfunction 0.9 ± 0.7 vs 0.7 ± 0.6 p > 0.05. | Not reported. |

*(Continued)*

**Table 4.** (Continued)

| Study ID | Definition of pulmonary rehabilitation | Definition of sleep quality | Follow up and result | Concordance to pulmonary rehabilitation |
|---|---|---|---|---|
| Spielmanns 2022 | Physical exercise training session for 6 months via the Kaia COPD app which has an exercise training programme, breathing exercises and education programme. There were various daily whole-body exercises of 15–20 minutes. | Duration of sleep and sleep efficiency based on activity tracker POLAR A370 watch. | Duration of sleep at baseline: intervention 7.61 ± 1.36 vs control 7.73 ± 1.08, p = 0.703. Duration of sleep at 6 months: intervention 7.60 ± 1.31 vs control 7.13 ± 1.69, p = 0.294, effect size 0.311 95%CI -0.269 to 0.888. Sleep efficiency at baseline: intervention vs control 91.71 ± 3.20% vs 90.84 ± 3.33%, p = 0.321. Sleep efficiency at 6 months: intervention vs control 91.95 ± 2.28% vs 90.75 ± 2.93%, p = 0.12, effect size 0.465 95%CI -0.120 to 1.046. | 5 out of 67 withdrew informed consent. |
| Thapamagar 2021 | Exercise training, extensive self-management educational program for 3 days as week for 8 weeks with 30–60 minutes of education, 2–2.5 hours of exercise and the use of Actiwatch 2 for 24 hours a day monitoring before and after programme. | PSQI and a global score of ≥5 was "poor" quality sleepers. Epworth sleepiness scale. Sleep efficiency, wakefulness after sleep onset and number of nightly awakening events based on actigraphy. | PSQI mean difference: Full cohort 0.57 95%CI -0.05 to 1.2, p = 0.07, normal sleep before (n = 9) -0.44 95%CI -2.3 to 1.4, p = 0.059, poor sleep (n = 52) 0.79 95%CI 0.07 to 1.4, p = 0.03. Outcomes pre- and post-pulmonary rehabilitation: Sleep efficiency (%): pre-80.6 95%CI 73.0–84.9, post- 79.5 95%CI 72.5–85.9, p = 0.32. Wakefulness after sleep onset (min): pre- 51.3 95%CI 37.2–70.6, post- 56.3 95%CI 33.5–75.6, p = 0.70. Number of nightly awakening events pre- 33.4 95%CI 25.3–45, post- 35.5 95%CI 22.3–46.5, p = 0.77. Mean change in Epworth sleepiness scale -0.3 95%CI -1.4 to 0.7, p = 0.54. | 7 out of 90 patients dropped out of pulmonary rehabilitation programme. |
| Yu 2022 | Mindfulness behavioural intervention with progressive breathing training with MDT COPD management team, assessment of the cognitive ability of patients, cognitive intervention, behavioural intervention and respiratory muscle training for 12-weeks. | Sleep based on Chinese version of COPD Assessment Test scale where 0–10 was mild, 11–20 was moderate, 21–30 was severe and 31–40 was very severe. | After the 12-week intervention, the quality-of-life score for sleep in the study group was lower in the control group (p < 0.05). | – |

PSQI=Pittsburgh Sleep Quality Index, PR=pulmonary rehabilitation, COPD=chronic obstructive pulmonary disease.

### Scores and tests and their correlation with PSQI

Only one study evaluated the correlation between tests and scores and the PSQI. McDonnell et al reported that the COPD assessment test ($p < 0.01$), Hospital Anxiety and Depression Scale-anxiety score ($p < 0.01$), and Hospital Anxiety and Depression Scale depression score ($p < 0.01$) were significantly correlated with the PSQI but not with the incremental shuttle walk test ($p > 0.05$) [19].

### Concordance with pulmonary rehabilitation

Eight of the included studies reported patients who were not followed up or did not complete PR. Benzo et al reported that 45 patients out of 188 in the PR group did not have 3 months follow up assessments [27]. In one of the two studies by Cox et al. and Kittah et al., 14.6% of patients did not complete PR, whilst in the other, 34.6% were excluded [18,25]. In the study by Mak et al., only 16 out 98 patients completed the 12-month post-PR assessment [28]. McDonnell et al reported that 28 patients

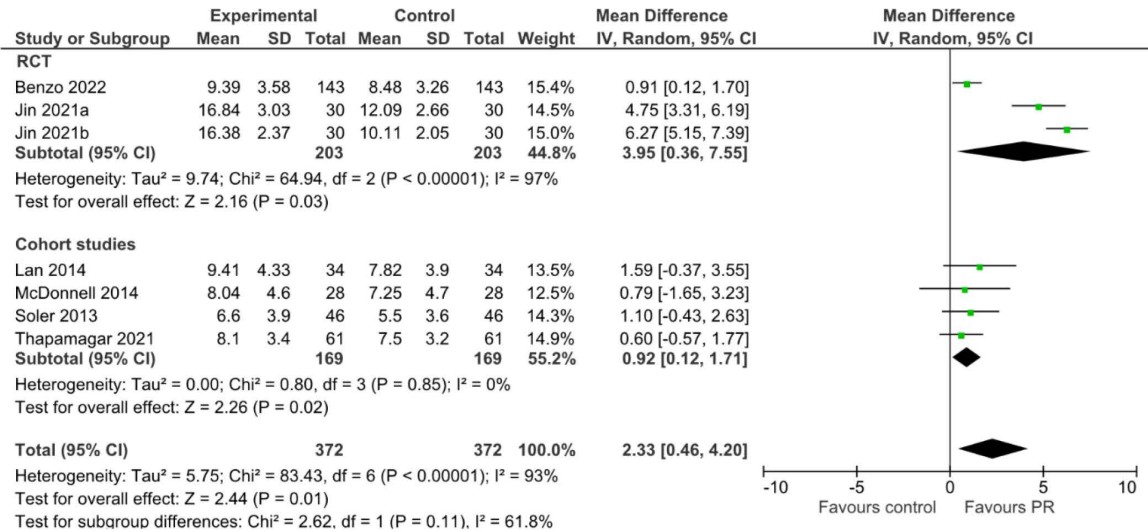

**Fig 2. Pooled analysis of PR on sleep quality as assessed by the PSQI.**

completed the first and second PSQI evaluation out of 61 eligible patients [19]. Shah et al reported that 2 patients were excluded because they were ill at the time of post-PR actigraphy [31]. A total of 7.5% of patients withdrew informed consent in the study by Spielmanns et al. [26] and 7.8% dropped out of the PR programme in the study by Thapamagar et al. [34].

## Discussion

This systematic review of the impact of PR on sleep quality has several key findings. Firstly, there is some evidence to suggest that PR has a beneficial effect on sleep quality in COPD patients as defined by the PSQI. Secondly, for other objective markers of sleep such as sleep onset latency, wake after sleep onset, total sleep time, and sleep efficiency sleep, there is no evidence to support the benefits of PR but more specific studies on these aspects are suggested. Thirdly, there is inconsistencies across the various PR programs, but they did follow the main principles which include aerobic exercise, upper limb exercise, lower limb exercise and resistance exercise, and were carried out for 7 weeks to 12 months with sessions lasting between 15min and 2 hours, 1 to 6 times a week. Fourthly, while the studies did evaluate patients with a diagnosis of COPD it is not possible to determine from them if PR is more beneficial for certain types of patients such as those with less or more severe COPD as there are many patients that do not complete with PR. Finally, the definition of sleep quality is inconsistent across the studies as some evaluated the PSQI, while others determined the Epworth Sleepiness Scale score, the Visual analogue scale, the Mini-sleep questionnaire, and markers from actigraphy or other non-invasive monitoring. Overall, these findings suggest that PR may be associated with improvements in sleep quality in patients with COPD and PR may be recommended for patients who feel that their sleep quality requires improvement.

The findings of this current study extend the knowledge about PR and sleep quality in COPD patients. The current findings support the findings of the meta-analysis by Kelley et al, which showed a significant improvement in overall sleep quality when treatment includes an exercise program [40]. The major differences between the two reviews are that this current review focused especially on patients with COPD, whereas the Kelley et al review did not restrict the cohort to specific patients, and that the Kelley et al review only included 3 meta-analyses comprising 950 patients compared to the current review which included 16 studies comprising 1478 patients with COPD. Kelley et al did suggest that there were statistical improvements in the apnea-hypopnea index, overall sleep quality, global score, subjective sleep, and sleep latency with exercise interventions. Another systematic review, carried out by Yang P. H. et al, focused on whether

exercise training programs improve the quality of sleep in middle-aged and older adults with sleep problems [41]. In their review of 6 randomized trials, they found moderate beneficial effects of exercise training on sleep quality, such as improvements in the Global PSQI scores and its sub-domains of subjective sleep quality, sleep latency and sleep medication use but no significant improvements were found regarding other sleep parameters such as sleep duration, efficiency and disturbance. A key consideration of their review compared to this current review is that in this review all patients in the trials are COPD patients whereas this is not the case in their review; moreover, their patients all previously had sleep problems. It is notable that improvements in sleep quality is more easily achieved among patients with poor sleep quality compared to patients which start with baseline good sleep quality.

PR is based on a comprehensive, multidisciplinary, patient-centred set of interventions, including exercise training, self-management, education and psychosocial support, that relieves dyspnoea and fatigue, improves emotional functioning, increases self-awareness of the disease, and contributes to improved health-related quality of life and exercise capacity [42]. For physiological reasons, PR further improves respiratory muscle strength by increasing aerobic enzymes to relax the muscles [43]. This improvement has been correlated with a reduction in dynamic hyperinflation [38] and improved ventilation in COPD patients. Moreover, PR also helps to improve systemic inflammation [44]. Pinho et al. (2007) concluded that patients with COPD are characterized by increased systemic and pulmonary oxidative stress and that PR is related to a reduction in oxidative stress [39]. Exercise training is also recommended to improve sleep quality by increasing energy expenditure, endorphin secretion and body temperature, thus promoting sleep to restore the body [45]. Physical health is said to be associated with sleep quality [46] and greater improvements in health status following exercise training are associated with better sleep outcomes [40]. Psychologically, as mentioned above, anxiety and depression can contribute to poor sleep quality in people with COPD and there is a high prevalence of depression and anxiety associated with COPD. PR led to better psychological status on the Anxiety-Depression Scale in patients with COPD [14]. Therefore, exercise training promotes sleep through its anxiolytic or antidepressant effects.

Our study highlights inconsistencies in the literature of what constitutes sleep quality. The PSQI was evaluated in 10 of the 16 studies as a marker of sleep quality. The PSQI is a self-reported questionnaire with 19 separate items, including open-ended questions and Likert-like items, with high test-retest reliability which is a validated assessment of sleep quality [47,48]. It includes sleep quality, sleep latency, sleep duration, habitual sleep efficiency, sleep disorders, sleep medication use, and daytime dysfunction. A Global PSQI score ≥5 is associated with poor sleep quality (diagnostic sensitivity 89.6%, specificity 86.5%) [42]. Two of 16 studies applied actigraphy which is a reliable sleep measurement and requires 6 consecutive nights to be accurate [49].

The clinical impact of this review is that its findings provide evidence for a positive association between PR and sleep quality in patients with COPD. These findings can support clinicians to recommend PR to patients with COPD who have poor quality sleep. The findings are also representative of the general population and can be generalized to many countries as the data were derived on an international basis from countries such as United States, China, Australia as well as from European countries.

Several recommendations for future randomised controlled trials appear to be justified based on the current findings. In particular, outstanding questions remain regarding what type of PR program is associated with the most significant improvements in sleep quality and whether certain types of patients benefit most from PR programs. A multi-armed randomised controlled trial with different PR regimes may be helpful to compare directly the dose-response effects of PR on selected sleep outcomes in COPD patients, including variations in the treatment arms for type of exercise, use of aerobic exercise, strength training or both. Second, whether PR is cost-effective in improving sleep outcomes in COPD patients requires further investigation as it is not well described in the literature. Finally, there is also scope for future studies to gain a better understanding of how to improve concordance to PR as the number of patients that dropped out of PR in previous trials or not followed up were not negligible. Future randomised controlled trials should report information on any adverse events experienced by participants during the intervention and reasons for drop-out or failure to complete PR or follow up.

## Limitations

There are several limitations within the current review. Firstly, six of the included studies were only available as conference abstracts, which lack detailed information regarding methodology and results. However, it is important to consider conference abstracts as inclusion of this grey literature can help reduce the potential for publication bias. Secondly, the average score for the quality of articles included was only 6.9, suggesting that the overall quality could be improved. Thirdly, there is significant methodology heterogeneity in the included studies as there is no consistent definition for PR in terms of time, activities, or periodicity of implementation. Finally, the measurement of sleep quality was variable, and the use of the PSQI could be considered subjective compared to more objective markers for evaluating sleep such as polysomnography.

## Conclusions

PR appears to be associated with improvements in sleep quality in patients with COPD. However, more studies are needed to determine what the ideal PR programme is that best improves sleep quality and whether certain patients benefit more compared to others.

## Supporting information

**S1 File.  GRADE evidence profile.**
(DOCX)

**S2 File.  Search results 25 Feb 2023 258.**
(DOC)

**S3 File.  Cochrane library search 3 Mar 2022.**
(DOCX)

**S4 File.  COPD pulmonary rehab systematic review 14 Mar 2023.**
(XLSX)

**S5 File.  Data analysis draft.**
(XLSX)

**S6 File.  PRISMA 2020 checklist.**
(DOCX)

## Author contributions

**Conceptualization:** Chun Shing Kwok.

**Data curation:** Shuiyan Dai.

**Formal analysis:** Chun Shing Kwok.

**Methodology:** Shuiyan Dai.

**Software:** Chun Shing Kwok.

**Supervision:** Chun Shing Kwok.

**Writing – original draft:** Shuiyan Dai.

**Writing – review & editing:** Chun Shing Kwok.

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
