## [Decision Letter · Decision Letter 0]

11 Mar 2024

PONE-D-23-38472The impact of pulmonary rehabilitation on sleep quality in patients with chronic obstructive pulmonary disease: A systematic review and meta-analysisPLOS ONE

Dear Dr. Dai,

Thank you for submitting your manuscript to PLOS ONE. After careful consideration, we feel that it has merit but does not fully meet PLOS ONE’s publication criteria as it currently stands. Therefore, we invite you to submit a revised version of the manuscript that addresses the points raised during the review process.

We look forward to receiving your revised manuscript.

Kind regards,

Alison Wang

Academic Editor

PLOS ONE

Journal Requirements:

2. PLOS requires an ORCID iD for the corresponding author in Editorial Manager on papers submitted after December 6th, 2016. Please ensure that you have an ORCID iD and that it is validated in Editorial Manager. To do this, go to ‘Update my Information’ (in the upper left-hand corner of the main menu), and click on the Fetch/Validate link next to the ORCID field. This will take you to the ORCID site and allow you to create a new iD or authenticate a pre-existing iD in Editorial Manager. Please see the following video for instructions on linking an ORCID iD to your Editorial Manager account: "https://www.youtube.com/watch?v=_xcclfuvtxQ".

4. Please include the reference section of your manuscript.

Reviewers' comments:

Reviewer's Responses to Questions

**Comments to the Author**

1. Is the manuscript technically sound, and do the data support the conclusions?

Reviewer #1: Yes

Reviewer #2: Yes

2. Has the statistical analysis been performed appropriately and rigorously? 

Reviewer #1: No

Reviewer #2: Yes

3. Have the authors made all data underlying the findings in their manuscript fully available?

Reviewer #1: Yes

Reviewer #2: Yes

4. Is the manuscript presented in an intelligible fashion and written in standard English?

Reviewer #1: Yes

Reviewer #2: No

5. Review Comments to the Author

Reviewer #1: Thank you for inviting me to review this systematic review and meta-analysis which evaluate the impact of PR on sleep quality in COPD. The authors aligned with guidelines of the systematic review and register the review prior to conduct the study. The outcome indicated that PR is improving sleep quality in patients with COPD. I admire the effort here in conducting and writing the paper. However, there are some concerns need attention from the reviewer point of view.

- As author indicated the meta-analysis included RCT and non RCT (observational) studies. It is highly recommended by the Cochrane guidelines that RCT must be treated alone in the meta-analysis because the study designs are different. Pooled results must be presenting RCTs only.

- Heterogeneity checked via RevMan present 93%. That’s fine. Then the author removed one study. Then, the heterogeneity dropped to 0%! I am impressed that one study can represent all the heterogeneity in the pooled analysis. It acceptable in the guidelines to deal with small to moderate heterogeneity in the analysis because studies are different. I suggest sperate the analysis for example do the pooled analysis for RCT only. Then, authors can present the pooled analysis for other studies if they can. However, authors admired in the limitations “there is significant methodology heterogeneity in the included studies as there is no consistent definition for PR in terms of time, activities, or periodicity of 300 implementation.”

- For better presenting in the result, present one study designed with pooled result for one outcome for example RCT and PSQI

- Including results from conference abstract can be risky. I suggest focusing on the study that presents data (full text publications)

- The keywords in the search strategy are not clear! Did the authors only searched these three terms. What bout Mesh terms? Any language filters?

- Author mentioned “we planned to perform an asymmetry test to determine 113 whether there was publication bias” Can not find the funnel plot in the figures? Or the sensitivity analysis?

- Quality assessment is not clear. Did the author used Ottawa-Newcastle Scale for all studies including the RCT? What about Cochrane risk of bias tool for RCTs?

- It would be better to present the data as systematic review and meta-analysis the main outcome PSQI.

- In the abstract as well as in the methods. It is random effect model not “Random effects meta-analysis.”

Thank you.

Reviewer #2: This is a creative topic, and the authors had done a good reviewing work. However, the manuscript should undergo major revision to meet the high standard basic requirements of the Journal. Here are some suggestions to further enhance the manuscript:

1. In the Introduction section, focus on the sleep quality issues among COPD patients and specifically detailing the positive and/or negative impacts of sleep quality on their quality of life.

2. In the introduction section, clarify the association of PR and sleep quality in COPD patients or the potential mechanism of PR on sleep quality. More details need to be specified on the knowledge gaps.

3. In the Methods section, clarify the PICO principle and list the inclusion criteria and exclusion criteria more specifically.

4. Clarify the reason of the PSQI as the primary outcomes on sleep quality in the methods section.

5. Clarify the actions if the two reviewers (SD and CSK) have a disagreement while engaging in study selection, data extraction and quality assessment?

6. Clarify the definition/criteria of each study quality for the NOS, e.g., NOS scores less than or equal to 4 points as low quality? Or how to identify the overall quality of 6.9 scores as the low or moderate or high quality.

7. In the data synthesis, how about the subgroup analysis. Specify the reason of using the random-effects meta-analysis.

8. In the results section, the contents of the PRISMA flow diagram were not completed. Please use the latest PRISMA 2020 flow diagram and provide sufficient information of the literature screening process.

9. The APA suggest " p value" The p is lowercase and italicized. Please re-check the manuscript.

10. In the discussion, use a cautious or tentative tone when describing or reporting the interpretation and explanation of the results.

11. Lacking note on abbreviation in the Table 2 and Table 3.

12. Reformat the reference style.

13. The manuscript should undergo careful language editing for spelling, grammar errors and be re-checked for any errors to meet the high standard basic requirements of the Journal.

6. PLOS authors have the option to publish the peer review history of their article (what does this mean? ). If published, this will include your full peer review and any attached files.

**Do you want your identity to be public for this peer review?** For information about this choice, including consent withdrawal, please see our Privacy Policy .

Reviewer #1: No

Reviewer #2: **Yes: ** Mengyuan Li

---

## [Author Response · Author response to Decision Letter 1]

25 Jun 2024

23 May 2024

Dr. Alison Wang

Academic Editor

PLOS One

Dear Dr. Wang,

Please find enclosed our manuscript entitled “The impact of pulmonary rehabilitation on sleep quality in patients with chronic obstructive pulmonary disease: A systematic review and meta-analysis” for consideration for publication in PLOS One.

Firstly, may we thank the Reviewers for their careful appraisal our paper and their detailed points, to which we respond to below. We feel that the reviewer’s comments have significantly improved the quality of our manuscript. We have attempted to answer all reviewer’s comments fully.

We hope that we have addressed all reviewers’ comments sufficiently and hope that these changes will enable publication of our paper in your journal.

Signed on behalf of submitting authors,

Ms Shuiyan Dai and Dr Chun Shing Kwok, MBBS PhD MSc BSc MRCP

Nurse and Senior Lecturer in Research Methodology and Cardiology Registrar

Reviewer #1: Thank you for inviting me to review this systematic review and meta-analysis which evaluate the impact of PR on sleep quality in COPD. The authors aligned with guidelines of the systematic review and register the review prior to conduct the study. The outcome indicated that PR is improving sleep quality in patients with COPD. I admire the effort here in conducting and writing the paper. However, there are some concerns need attention from the reviewer point of view.

- As author indicated the meta-analysis included RCT and non RCT (observational) studies. It is highly recommended by the Cochrane guidelines that RCT must be treated alone in the meta-analysis because the study designs are different. Pooled results must be presenting RCTs only.

Response: We agree with the review that RCTs provide the highest quality of evidence which is superior to observational studies. However, trials are limited that they may not include real-world patients encountered in everyday practice. We do agree that it is important to highlight the findings from RCT and cohort studies so we have stratified the analysis. This is important as we feel that a review should include all the evidence not just those of higher quality so that we can fairly represent all the literature before making conclusions. The Figure 2 is now revised as follows:

In the methods, we have added the following:

“The main analysis which included studies that were of randomized controlled trial (RCT) and cohort in design was stratified by study design so that the pooled findings from RCT could be distinguished from cohort studies.”

In the results we have added the following:

“The benefit of PR was observed to a greater extent among studies that were of RCT design (mean difference 3.95 95% CI 0.36 to 7.55, I2=97%, 2 studies) compared to cohort studies (mean difference 0.92 95% CI 0.12 to 1.71, I2=0%, 4 studies).”

- Heterogeneity checked via RevMan present 93%. That's fine. Then the author removed one study. Then, the heterogeneity dropped to 0%! I am impressed that one study can represent all the heterogeneity in the pooled analysis. It acceptable in the guidelines to deal with small to moderate heterogeneity in the analysis because studies are different. I suggest sperate the analysis for example do the pooled analysis for RCT only. Then, authors can present the pooled analysis for other studies if they can. However, authors admired in the limitations "there is significant methodology heterogeneity in the included studies as there is no consistent definition for PR in terms of time, activities, or periodicity of 300 implementation."

Response: We have stratified the analysis by RCT and non-RCT and made the RCT subgroup analysis the primary outcome as shown above.

- For better presenting in the result, present one study designed with pooled result for one outcome for example RCT and PSQI

Response: We have stratified the analysis for RCT and non-RCT but kept the overall pooled results across all studies so the collective effect can be examined as well as within the subgroups.

- Including results from conference abstract can be risky. I suggest focusing on the study that presents data (full text publications)

Response: We included conference abstracts in this manuscript because inclusion of grey literature would reduce the risk of publication bias.

- The keywords in the search strategy are not clear! Did the authors only searched these three terms. What bout Mesh terms? Any language filters?

Response: We used a broad search using few terms in order to maximize sensitivity (including more results to reduce the risk of missing studies) as oppose to a narrow search with many search terms including Mesh terms (this strategy often results in fewer potential studies so that there are fewer studies to screen thus more specific). If the broad terms of sleep quality and pulmonary rehabilitation were used then it would also encompass the Mesh terms and therefore the Mesh search was not done. There were not language filters to reduce the chance of missing relevant studies.

- Author mentioned "we planned to perform an asymmetry test to determine 113 whether there was publication bias" Can not find the funnel plot in the figures? Or the sensitivity analysis?

Response: It is stated in our methods:

“If there were more than 10 studies in the meta-analysis and the statistical heterogeneity was less than 50%, we planned to perform an asymmetry test to determine whether there was publication bias.”

None of the pooled analyses had greater than 10 studies so asymmetry testing was not performed. The sensitivity analysis was performed to explore statistical heterogeneity. The results of the sensitivity analysis is stated as follows in the methods:

“However, the source of the heterogeneity was the study by Jin et al. If this study is removed the estimate would still be significantly different but the effect size would be smaller with no statistical heterogeneity (mean difference 0.91 95%CI 0.35-1.48, I2=0%, 5 studies).”

- Quality assessment is not clear. Did the author used Ottawa-Newcastle Scale for all studies including the RCT? What about Cochrane risk of bias tool for RCTs?

Response: In order to maintain consistency of the risk of bias assessment we used the Ottawa-Newcastle Scale. Many of the criteria used to assess quality such as ascertainment of exposure and outcome apply to both cohort studies and RCT. The consistency was important so that we can consider quality across all of the studies.

- It would be better to present the data as systematic review and meta-analysis the main outcome PSQI.

Response: We have added to the methods:

“The primary outcome of this study was the Pittsburgh Sleep Quality Index (PSQI) and the mean and standard deviation in the Pittsburgh Sleep Quality Index (PSQI) before and after PR were pooled.”

- In the abstract as well as in the methods. It is random effect model not "Random effects meta-analysis."

Response: Thank you for drawing our attention to this error. It is now corrected as follows in the abstract:

“Meta-analysis with the random effects model was performed to determine if PR is associated with any difference in the Pittsburgh Sleep Quality Index (PSQI).”

And in the methods:

“RevMan 5.4 (The Nordic Cochrane Centre, The Cochrane Collaboration, Copenhagen, Denmark) was employed in performing meta-analysis with the random effects model using the mean difference method.”

Thank you.

Response: Thank you for taking the time to review our manuscript.

Reviewer #2: This is a creative topic, and the authors had done a good reviewing work. However, the manuscript should undergo major revision to meet the high standard basic requirements of the Journal. Here are some suggestions to further enhance the manuscript:

1. In the Introduction section, focus on the sleep quality issues among COPD patients and specifically detailing the positive and/or negative impacts of sleep quality on their quality of life.

Response: We have revised the text in the introduction as follows:

“Sleep quality is poor in patients with severe COPD compared with normative populations of similar age, and daytime hypoxaemia is independently associated with impaired sleep efficiency. Approximately 70% of COPD patients typically complain of difficulty sleeping, two to three times more than the general population. One study suggests that 39% of patients with nocturnal cough or wheeze report difficulty initiating or maintaining sleep. Patients with COPD may experience anxiety, depression, shortness of breath and coughing, sputum, and hypoxaemia during the night. There is also some evidence to suggest that patients with COPD may be affected by an overlap with obstructive sleep apnoea as the prevalence of obstructive sleep apnoea ranges from 10 to 65%. There appears to be a bidirectional relationship between sleep quality and clinical outcomes, with sleep disturbances leading to systemic inflammation, immune impairment, lack of exercise and cognitive changes, which may affect medication adherence and lead to poor clinical outcomes, nocturnal COPD symptoms, and reduced physical activity. Sleep disturbance in patients with COPD likely contributes to the non-specific daytime symptoms of chronic fatigue, lethargy and overall impairment in quality of life.

2. In the introduction section, clarify the association of PR and sleep quality in COPD patients or the potential mechanism of PR on sleep quality. More details need to be specified on the knowledge gaps.

Response: Thank you for the suggestion. The information on mechanism for PR and sleep quality is in the discussion currently because we feel that the introduction text is long enough. If the reviewer feels that it is better placed there then we can move it there but the introduction will become long. The text in the discussion is written as follows:

“PR is based on a comprehensive, multidisciplinary, patient-centred interventions, including exercise training, self-management, education and psychosocial support, relieves dyspnoea and fatigue, improves emotional functioning, increases self-awareness of the disease, and contributes to improved health-related quality of life and exercise capacity. For physiological reasons, PR further improves respiratory muscle strength by increasing aerobic enzymes to relax the muscles. This improvement has been correlated with a reduction in dynamic hyperinflation and improved ventilation in COPD patients. Moreover, PR also helps to improve systemic inflammation. Pinho et al concluded that patients with COPD are characterized by increased systemic and pulmonary oxidative stress and that PR is related to a reduction in oxidative stress. Exercise training is also recommended to improve sleep quality by increasing energy expenditure, endorphin secretion and body temperature, thus promoting sleep to restore the body. Physical health is said to be associated with sleep quality and greater improvements in health status following exercise training are associated with better sleep outcomes. Psychologically, as mentioned above anxiety and depression can contribute to poor sleep quality in people with COPD and there is a high prevalence of depression and anxiety in COPD. PR led to better psychological status on the Anxiety-Depression Scale in patients with COPD. Therefore, exercise training promotes sleep through its anxiolytic or antidepressant effects.”

3. In the Methods section, clarify the PICO principle and list the inclusion criteria and exclusion criteria more specifically.

Response: We have placed the inclusion criteria within the PICO framework as suggested as follows:

“The participants were patients with a diagnosis of COPD. The intervention is pulmonary rehabilitation and the comparator is the same patient prior to taking part in pulmonary rehabilitation. The outcome is sleep quality.”

4. Clarify the reason of the PSQI as the primary outcomes on sleep quality in the methods section.

Response: We have added the following text in the methods manuscript:

“The PSQI was chosen because it is the most commonly used generic measure of sleep quality in clinical and research settings.”

5. Clarify the actions if the two reviewers (SD and CSK) have a disagreement while engaging in study selection, data extraction and quality assessment?

Response: We have added the following text in the methods manuscript:

“Where there were disagreements in study selection, data extraction and quality assessment between the two reviewers, the third reviewer (SS) was consulted and a decision was made by consensus.”

6. Clarify the definition/criteria of each study quality for the NOS, e.g., NOS scores less than or equal to 4 points as low quality? Or how to identify the overall quality of 6.9 scores as the low or moderate or high quality.

Response: The NOS score was present for each study which has an integer value from 0 to 9 where lower quality studies are represented by lower values while higher values represent higher quality studies. We describe the overall quality of the studies as the average score across all included studies.

7. In the data synthesis, how about the subgroup analysis. Specify the reason of using the random-effects meta-analysis.

Response: We have performed subgroup analysis according to study design. This is shown below. The random effects meta-analysis was used because we could not be certain that there is homogeneity among the methodology of the different studies which would support fixed-effects meta-analysis.

8. In the results section, the contents of the PRISMA flow diagram were not completed. Please use the latest PRISMA 2020 flow diagram and provide sufficient information of the literature screening process.

Response: The flow diagram is shown below based on the PRISMA 2020 flow diagram. We were able to retrieve all studies that were potentially relevant in the screening process so the box for studies which were not retrieved was not included.

9. The APA suggest " p value" The p is lowercase and italicized. Please re-check the manuscript.

Response: We have revised as suggested.

10. In the discussion, use a cautious or tentative tone when describing or reporting the interpretation and explanation of the results.

Response: We have revised the text as suggested as follows:

“Firstly, there is some evidence to suggest that PR has a beneficial effect on sleep quality in COPD patients as defined by the PSQI.”

“Overall, these findings suggest that PR may be associated with improvements in sleep quality in patients with COPD and PR may be recommended for patients who feel that their sleep quality requires improvement.”

We would welcome any additional specific changes the reviewer can suggest.

11. Lacking note on abbreviation in the Table 2 and Table 3.

Response: We have explained the abbreviations as follows:

Table 2

COPD=chronic obstructive pulmonary disease, PSQI=Pittsburgh Sleep Quality Index, PR=pulmonary rehabilitation, MDT=multidisciplinary team

Table 3

PSQI=Pittsburgh Sleep Quality Index, PR=pulmonary rehabilitation, COPD=chronic obstructive pulmonary disease

12. Reformat the reference style.

Response: We have revised as suggested.

13. The manuscript should undergo careful language editing for spelling, grammar errors and be re-checked for any errors to meet the high standard basic requirements of the Journal.

Response: We have checked and would welcome any suggestions on improving the text of the manuscript.

14.We note that your Data Availability Statement is currently as follows: [All relevant data are within the manuscript and its Supporting Information files.]

Response: We do not have permission. This is a review and we do not have the original data. We can be obtained from the individual studies.

---

## [Decision Letter · Decision Letter 1]

29 Aug 2024

PONE-D-23-38472R1The impact of pulmonary rehabilitation on sleep quality in patients with chronic obstructive pulmonary disease: A systematic review and meta-analysisPLOS ONE

Dear Dr. Dai,

Thank you for submitting your manuscript to PLOS ONE. After careful consideration, we feel that it has merit but does not fully meet PLOS ONE’s publication criteria as it currently stands. Therefore, we invite you to submit a revised version of the manuscript that addresses the points raised during the review process.

We look forward to receiving your revised manuscript.

Kind regards,

Pasquale Tondo, MD

Academic Editor

PLOS ONE

Reviewers' comments:

Reviewer's Responses to Questions

**Comments to the Author**

1. If the authors have adequately addressed your comments raised in a previous round of review and you feel that this manuscript is now acceptable for publication, you may indicate that here to bypass the “Comments to the Author” section, enter your conflict of interest statement in the “Confidential to Editor” section, and submit your "Accept" recommendation.

Reviewer #2: (No Response)

Reviewer #3: (No Response)

2. Is the manuscript technically sound, and do the data support the conclusions?

Reviewer #2: Partly

Reviewer #3: Yes

3. Has the statistical analysis been performed appropriately and rigorously? 

Reviewer #2: Yes

Reviewer #3: Yes

4. Have the authors made all data underlying the findings in their manuscript fully available?

Reviewer #2: No

Reviewer #3: Yes

5. Is the manuscript presented in an intelligible fashion and written in standard English?

Reviewer #2: Yes

Reviewer #3: Yes

6. Review Comments to the Author

Reviewer #2: I admire the effort here in conducting such work. However, there are still some concerns were not solved clearly and need further attention from the reviewer point.

1.The definition/description of the Comparator(s)/control in your paper was misunderstanding and did not keep consistent with your registered protocol.

2.For the search strategy, I recommend to follow the JBI methodology and PRISMA reporting guideline using the appropriate search strategy. Only these three terms didn’t mean a broader search. How about the articles that may use different words (e.g., synonyms and variations) to describe the same concept. To ensure the sensitivity and accuracy during the retrieval, also the broader search, that is why we need a combination of mesh terms and free/key words, along with the use of appropriate search queries. This can enhance efficiency, accuracy, and comprehensiveness of searches in medical and scientific literature databases. Additionally, describe at least one database that utilizes a search strategy enabling others to replicate your retrieval results.

3.In the section of Study inclusion and exclusion criteria, it’s said “Studies had to have original data which meant that letters, editorials, comments, and reviews were excluded.” However, this paper included the abstracts WITHOUT original data. If the abstracts were included as you explained “the grey literature would reduce the risk of publication bias ” in the response letter, then you have to search the grey literature databases to minimize the publication bias as much as possible, but you did not. These statements are mutually contradictory.

4.The Newcastle Ottawa Scale (NOS) recommended by the Cochrane Collaboration was used as a quality assessment scale for case-control and cohort studies but NOT for randomized studies. Using the Newcastle Ottawa Scale (NOS) for RCT’s quality assessment seems inappropriate, iI suggest to use the specific quality assessment tools for RCT.

5.Refer to the results of quality assessment, we did not see how many papers were identified as high or low or moderate quality. As the quality of the included studies would impact the robustness of conclusions. There is indeed a cut-off score of NOS.

Reviewer #3: (No Response)

7. PLOS authors have the option to publish the peer review history of their article (what does this mean? ). If published, this will include your full peer review and any attached files.

**Do you want your identity to be public for this peer review?** For information about this choice, including consent withdrawal, please see our Privacy Policy .

Reviewer #2: No

Reviewer #3: No

---

## [Author Response · Author response to Decision Letter 2]

9 Nov 2024

14 September 2024

Dr. Alison Wang

Academic Editor

PLOS One

Dear Dr. Wang,

Please find enclosed our manuscript entitled “The impact of pulmonary rehabilitation on sleep quality in patients with chronic obstructive pulmonary disease: A systematic review and meta-analysis” for consideration for publication in PLOS One.

Firstly, may we thank the Reviewers for their careful appraisal our paper and their detailed points, to which we respond to below. We feel that the reviewer’s comments have significantly improved the quality of our manuscript. We have attempted to answer all reviewer’s comments fully.

We hope that we have addressed all reviewers’ comments sufficiently and hope that these changes will enable publication of our paper in your journal.

Signed on behalf of submitting authors,

Ms Shuiyan Dai and Dr Chun Shing Kwok, MBBS PhD MSc BSc MRCP

Nurse and Senior Lecturer in Research Methodology and Consultant Cardiologist

Reviewer's Responses to Questions

Comments to the Author

1. If the authors have adequately addressed your comments raised in a previous round of review and you feel that this manuscript is now acceptable for publication, you may indicate that here to bypass the "Comments to the Author" section, enter your conflict of interest statement in the "Confidential to Editor" section, and submit your "Accept" recommendation.

Reviewer #2: (No Response)

Reviewer #3: (No Response)

Response: We hope the following response are satisfactory to address your comments.

2. Is the manuscript technically sound, and do the data support the conclusions?

Reviewer #2: Partly

Reviewer #3: Yes

Response: We hope the following response are satisfactory to address your comments.

3. Has the statistical analysis been performed appropriately and rigorously?

Reviewer #2: Yes

Reviewer #3: Yes

Response: Thank you.

4. Have the authors made all data underlying the findings in their manuscript fully available?

Reviewer #2: No

Reviewer #3: Yes

Response: There is no original data in this manuscript as it is a systematic review. All the data is available from original sources and we do not have permission to make the fully available other than seeking them directly from authors and journals.

5. Is the manuscript presented in an intelligible fashion and written in standard English?

Reviewer #2: Yes

Reviewer #3: Yes

Response: Thank you.

6. Review Comments to the Author

Reviewer #2: I admire the effort here in conducting such work. However, there are still some concerns were not solved clearly and need further attention from the reviewer point.

1.The definition/description of the Comparator(s)/control in your paper was misunderstanding and did not keep consistent with your registered protocol.

Response: We do not mention comparator or control group so there is no misunderstanding or inconsistency as there is no restriction based on comparator group. We have revised the text as follows:

“Studies were included if they evaluated sleep quality in people with COPD who underwent pulmonary rehabilitation and there was no restriction based on the presence or type of comparator group.”

2. For the search strategy, I recommend to follow the JBI methodology and PRISMA reporting guideline using the appropriate search strategy. Only these three terms didn't mean a broader search. How about the articles that may use different words (e.g., synonyms and variations) to describe the same concept. To ensure the sensitivity and accuracy during the retrieval, also the broader search, that is why we need a combination of mesh terms and free/key words, along with the use of appropriate search queries. This can enhance efficiency, accuracy, and comprehensiveness of searches in medical and scientific literature databases. Additionally, describe at least one database that utilizes a search strategy enabling others to replicate your retrieval results.

Response: We understand the approach that many organizations promote but the senior author who has extensive research experience who has used this approach before in a Cochrane review does not agree with this approach. We believe that in the current work additional terms will not add anything that is not captured in the parent term. The current approach was what was registered with PROSPERO and we should not deviate from it.

However, if the reviewer insists on having a new search, please suggest a strategy and we will try to accommodate it.

3. In the section of Study inclusion and exclusion criteria, it's said "Studies had to have original data which meant that letters, editorials, comments, and reviews were excluded." However, this paper included the abstracts WITHOUT original data. If the abstracts were included as you explained "the grey literature would reduce the risk of publication bias " in the response letter, then you have to search the grey literature databases to minimize the publication bias as much as possible, but you did not. These statements are mutually contradictory.

Response: I think there is misunderstanding about what original data is. Abstracts contain original data as they may report studies which collect data from patients. Their reporting may not be as complete as full manuscripts and they may not undergo the same level of peer review. If we excluded abstracts which may contain less publishable results then there may be publication bias and inclusion of this grey literature is important. Therefore, the statements are not mutually contradictory. We have added the following to clarify:

“Studies had to have original data which could be in full manuscript or abstract form so letters, editorials, comments, and reviews without original data were excluded.”

4. The Newcastle Ottawa Scale (NOS) recommended by the Cochrane Collaboration was used as a quality assessment scale for case-control and cohort studies but NOT for randomized studies. Using the Newcastle Ottawa Scale (NOS) for RCT's quality assessment seems inappropriate, iI suggest to use the specific quality assessment tools for RCT.

Response: We agree with the reviewer that the NOS is for observational studies and not designed for RCT but we used a single risk assessment for consistency. However we have now revised the risk assessment for trials to one recommended by Cochrane. Results are on the next page and we have added the following test in the methods and results:

“Study quality assessments for observational studies were carried out using the Ottawa-Newcastle Scale for observational studies [25].”

“The study quality assessment for randomised trials was performed using the Cochrane Risk of Bias Tool [26]. The domains assessed were bias from randomization process, bias due to deviations from intended intervention, bias in measurement of the outcome, and bias in selection of the reported result.”

“Among the five randomized controlled trials included in the review, two studies were classified as low risk for all five domains of the Cochrane risk of bias tool.”

Table 3: Study Quality Assessment by Cochrane Risk of Bias Tool for Randomised trials

Study ID Bias from randomization process Bias due to deviations from intended interventions Bias due to missing outcome data Bias in measurement of the outcome Bias in selection of the reported result

Benzo 2022 Some concern, randomized in a 1:1 ratio on the basis of a pregenerated sequence of assignments

through a computer-generated permuted

block randomization with blocks of size four, allocation concealment unclear. Low risk, 12-week program with weekly healthcare calls and a remote monitoring system, Garmin Vivofit activity monitor and oximeter vs usual care and educational packet

for weekly self-study. Low risk, participants in both arms did not complete measures at 3 months, lost contact and refused participation. Some patient died and became ineligible in usual care group. Some concern, assessors were not blinded to intervention received. Low risk, trial analysed according with pre-specified plan.

Cox 2019 Low risk, randomisation undertaken using a computer-generated sequence and allocation will be concealed using sealed, opaque envelopes. Low risk, home versus hospital-based rehabilitation program.

Low risk, no missing outcome data reported. Low risk, Objective measures of sleep quality were obtained from the SenseWear Armband. Low risk, trial analysed according with pre-specified plan although this was post-hoc analysis.

Jin 2021 Some concern, random number table used for randomisation but allocation concealment unclear. Low risk, pulmonary rehabilitation training three times a week in addition to conventional drug treatment vs drug treatment alone.

Low risk, no missing outcome data reported. Some concern, participants and assessors were not blinded. Low risk, trial analysed according with pre-specified plan.

Spielmanns 2022 Low risk, randomisation using a software randomizer and fabrication of lists done by the sponsor for allocation concealment. Low risk, KAIA COPD app program vs usual care. Low risk, only 6 out of 67 withdrew informed consent or discontinued by investigators. Low risk, electronic patient-reported outcomes and activity and sleep measured passively and continuously for each participant by the activity tracker. Low risk, trial analysed according with pre-specified plan.

Yu 2022 Some concern, randomization by random number table but allocation concealment unclear. Low risk, mindfulness behavioural intervention combined with progressive breathing training vs symptomatic treatment during hospitalization and routine nursing measures in internal medicine. Low risk, no missing outcome data reported. Some concern, participants and assessors were not blinded. Low risk, trial analysed according with pre-specified plan.

5. Refer to the results of quality assessment, we did not see how many papers were identified as high or low or moderate quality. As the quality of the included studies would impact the robustness of conclusions. There is indeed a cut-off score of NOS.

Response: The cutoffs for number of stars are arbitrary as there are 9 possible stars and it should sort of depend on the research question and the extent to which study designs can best answer the question. In this case we can assign high quality of 7-9 stars, 4-6 stars as moderate quality and 0-3 stars as low quality. We have added the following:

“Among the eleven observational studies, six had seven out of nine stars and would be considered as high quality compared to the other five studies which had six out of nine stars based on the Ottawa-Newcastle Scale.”

Reviewer #3: (No Response)

Response: Thank you.

---

## [Decision Letter · Decision Letter 2]

16 Jan 2025

The impact of pulmonary rehabilitation on sleep quality in patients with chronic obstructive pulmonary disease: A systematic review and meta-analysis

PONE-D-23-38472R2

Dear Dr. Dai,

We’re pleased to inform you that your manuscript has been judged scientifically suitable for publication and will be formally accepted for publication once it meets all outstanding technical requirements.

Kind regards,

Pasquale Tondo, MD

Academic Editor

PLOS ONE

Additional Editor Comments (optional):

Reviewers' comments:

Reviewer's Responses to Questions

**Comments to the Author**

1. If the authors have adequately addressed your comments raised in a previous round of review and you feel that this manuscript is now acceptable for publication, you may indicate that here to bypass the “Comments to the Author” section, enter your conflict of interest statement in the “Confidential to Editor” section, and submit your "Accept" recommendation.

Reviewer #2: (No Response)

Reviewer #3: All comments have been addressed

2. Is the manuscript technically sound, and do the data support the conclusions?

Reviewer #2: (No Response)

Reviewer #3: Yes

3. Has the statistical analysis been performed appropriately and rigorously? 

Reviewer #2: Yes

Reviewer #3: Yes

4. Have the authors made all data underlying the findings in their manuscript fully available?

Reviewer #2: Yes

Reviewer #3: Yes

5. Is the manuscript presented in an intelligible fashion and written in standard English?

Reviewer #2: Yes

Reviewer #3: Yes

6. Review Comments to the Author

Reviewer #2: (No Response)

Reviewer #3: From my point of view, the authors have addressed all relevant remarks raised in the review process.

7. PLOS authors have the option to publish the peer review history of their article (what does this mean? ). If published, this will include your full peer review and any attached files.

**Do you want your identity to be public for this peer review?** For information about this choice, including consent withdrawal, please see our Privacy Policy .

Reviewer #2: No

Reviewer #3: No

---

## [Editor Report · Acceptance letter]

PONE-D-23-38472R2

PLOS ONE

Dear Dr. Dai,

I'm pleased to inform you that your manuscript has been deemed suitable for publication in PLOS ONE. Congratulations! Your manuscript is now being handed over to our production team.

Kind regards,

on behalf of

Dr. Pasquale Tondo

Academic Editor

PLOS ONE